# Patient Education and Continuing Medical Education to Promote Shared Decision-Making. A Systematic Literature Review

**DOI:** 10.3390/ijerph16142482

**Published:** 2019-07-12

**Authors:** Anke Wagner, Natalia Radionova, Monika A. Rieger, Achim Siegel

**Affiliations:** Institute of Occupational and Social Medicine and Health Service Research, University Hospital Tübingen, Wilhelmstraße 27, 72074 Tübingen, Germany

**Keywords:** shared decision-making, patient involvement, patient education, continuing medical education, literature review

## Abstract

*Background*: Over recent years, the use of decision aids to promote shared decision-making have been examined. Studies on patient education and on continuing medical education for physicians are less common. This review analyzes intervention and evaluation studies on patient education and continuing medical education which aim to enhance shared decision-making. The following study parameters are of interest: Study designs, objectives, numbers of participants in the education courses, interventions, primary results, and quality of the studies. *Methods*: We systematically searched for suitable studies in two databases (Pubmed and the Cochrane Database of Systematic Reviews) from the beginning of April through to mid-June 2016. *Results*: 16 studies from a total of 462 hits were included: Three studies on patient education and 13 studies on continuing medical education for physicians. Overall, the study parameters were heterogeneous. Major differences were found between the courses; how the courses were conducted, their length, and participants. *Conclusions*: The differences found in the studies made it difficult to compare the interventions and the results. There is a need for studies that systematically evaluate and further develop interventions in this area to promote shared decision-making.

## 1. Introduction

Shared decision-making describes an interaction process with the aim of patients and physicians coming to jointly-made decisions based on their equal and active participation using shared information [1]. Shared decision-making represents a so-called middle position between the paternalistic model and the information model [2]. According to Charles et al., shared decision-making is characterized by the following key elements: (1) it involves physician and patient, (2) both parties share information, (3) both parties are in a process to build a consensus about the preferred treatment, and (4) both sides will decide jointly which treatment will be implemented [2].

In order to promote shared decision-making between patients and practitioners in clinical practice, the following three strategies have been applied in previous years [3,4,5].

Patient education: Patient education should prepare patients for greater involvement, both by fostering a general increase of patient competence (empowerment) and by improving communication skills for communicating with physicians [3,4,5,6,7,8]. Following patient education, patients should be capable of asking more questions during a consultation and expressing a greater desire for involvement [3]. 

Continuing medical education: In continuing medical education, physicians receive communication training and are educated on how to implement shared decision-making in their daily practice [3,5,7,8]. A high level of communication skills is required of medical professionals in order to inquire about patients’ preferences regarding participation in medical decisions. Physicians should be able to communicate disadvantages, risks, and uncertainty, sensitively and neutrally to patients. This requires specific discussion and action skills, which may need to be practiced [5]. Within the framework of further medical training, measures for shared decision-making, specific medical discussion and action skills are imparted. Based on well evaluated advanced training measures that are also in place in Germany, appropriate training processes for physicians were developed and examined. 

Decision aids: Decision aids include an understandable presentation of the treatment options with their respective advantages and disadvantages as well as with information on the probability of treatment success [3,5]. In previous years, criteria for their development and evaluation were developed for quality assurance (International Patient Decision Aid Standards) [3].

Closer consideration of the currently-available studies shows that examination of the use of decision aids has predominated. There are correspondingly fewer studies on patient education and continuing medical education. The following systematic review consequently concentrated on identifying and analyzing studies on patient education courses and continuing medical education which aim to enhance shared decision-making. 

## 2. Materials and Methods 

### 2.1. Search Strategy

From the beginning of April through to mid-June 2016, a systematic review of the literature in the databases Pubmed and the Cochrane Database of Systematic Reviews was carried out. In order to find studies for our specific research aim, we performed a broad-based search using the following keywords: “Patient participation” (MeSH) OR “Patient Education as Topic” (MeSH), “Patient participation” (MeSH) OR “training”, “Patient participation” OR “Patient Education”.

### 2.2. Inclusion Criteria

We included (1) studies on patient education with the aim of advancing patients’ communication skills; (2) studies on continuing medical education, in which physicians received training on communication skills and how to implement shared decision-making; (3) randomized controlled studies (RCTs) and evaluation studies, in which patient education and continuing medical education were evaluated as interventions. The studies had to be published between 2006 and June 2016 and be in German and/or English language.

### 2.3. Exclusion Criteria

We excluded study protocols and publications with predominantly methodological content. We did not take into account studies published before 2006. Studies reporting in languages other than English or German were also excluded in this review.

### 2.4. Information Extraction and Quality Assessment

The results of the literature search were initially assessed independently of two reviewers (AW and NR) based on the titles and abstracts to determine applicability to the study question. In the event of discrepancies between the reviewers, a third reviewer (MR) was consulted to enable a final decision. In the case of a further inclusion in the literature review, a full-text analysis was carried out. The two reviewers (AW and NR) read the studies and independently extracted content of interest. The following components were of interest when analyzing the studies: (1) Study designs and objectives, (2) number of course participants, (3) implemented interventions, (4) main study results, and (5) study quality. For the quality assessment, the internal validity of the selected studies was evaluated by two of the authors (AW and NR) using the SURE criteria. For studies with an interventional design, we applied the SURE checklist for randomized controlled trials and other experimental studies [9]. For observational studies, we used the SURE checklist for cohort studies [10].

## 3. Results

The database searches resulted in a total of 462 hits. After excluding duplicates (*n* = 55), 407 publications were first evaluated independently by two reviewers (AW and NR). After the screening phase, 91 hits were available for full text analysis. Following the full text analysis, a total of 16 studies were included in the literature review. Of the 16 studies, three studies address patient education as an intervention [11,12,13] and 13 studies deal with continuing medical education as an intervention [14,15,16,17,18,19,20,21,22,23,24,25,26]. The selection process for the studies is outlined in Figure 1.

### 3.1. Characteristics of Included Studies

Two studies were carried out in the United States of America [12,24] and one study is from United States of America/Puerto Rico [11]. Eight studies were carried out in Germany [13,15,16,17,18,22,23,25], three in Canada [19,20,21], and one in the Netherlands [26]. An additional, international multicenter study was carried out in Australia, New Zealand, Switzerland, Germany, and Austria [14] (see Table 1).

### 3.2. Studies on Patient Education

#### 3.2.1. Study Designs and Objectives 

The studies on patient education included two randomized controlled studies [11,13] and one before-and-after study [12]. The studies by Deen et al. and Alegria et al. evaluated trainings which aimed to enable patients to ask more questions during a consultation and to make decisions together with their physicians [11,12]. In the study by Hamann et al., the training aimed to foster communication skills particularly in patients with schizophrenic and schizoaffective disorders [13]. The following target parameters were recorded after the intervention:

One study measured patient activation using the PAM scale (Patient Activation Measure) [12].

One study recorded, among other things, desire for participation (Autonomy Preference Index), attitudes regarding medication (Beliefs in Medication Questionnaire), and satisfaction with their treatment [13].

Another study focused on patient activation (Patient Activation Scale) and self-management (Perceived Efficacy in Patient-Physician Interactions) [11].

#### 3.2.2. Number of Course Participants

The number of course participants in the individual studies ranged from 61 to 647 patients [11,12,13]. In two studies, patient age was an average of approximately 40 years [12,13]. The third study (Alegria et al.) presented the age range instead of an averaged age value. Patients between the ages of 18–70 years were included [11].

#### 3.2.3. Interventions

##### Persons Involved and Intervention Timing

The interventions were performed by various groups of people. In the study by Deen et al., an interviewer, whose professional affiliation was not mentioned, carried out the intervention [12]. The intervention took place immediately preceding a physician consult [12]. The two other studies evaluated made no reference to the timing of the interventions [11,13]. In the study by Harmann et al., the intervention was led by a psychiatrist and a psychologist [13]; in the study by Alegria et al., the intervention was performed by head nurses who were trained in advance during a two-day workshop [11].

##### Description of the Interventions

The interventions and the implemented methods were carried out differently in the studies (also see Table 2). The intervention in Alegria et al. consisted of three consecutive training sessions and was carried out in a patient group [11]. Brainstorming was used in these sessions and was reinforced using role play and practical tasks [11]. In the study by Deen et al., the intervention was carried out as a one-on-one interview [12]. For this purpose, a procedure was a utilized to help patients develop questions for the consultation through a brainstorming session, and to prioritize the questions according to personal relevance [12]. After the intervention, patients received the list of prioritized questions for use in the consultation [12]. The training in Hamann et al. was also conducted in a patient group and utilized role play and included behavior-oriented aspects [13]. Emphasis was on mutual support and interaction within the group [13].

##### Duration

The duration of the interventions was also described differently. There was no mention of intervention duration in the study by Deen et al. [12]. The interview intervention by Hamann et al. consisted of five one-hour sessions [13]. For their interview intervention, Alegria et al. implemented three training sessions of 30 to 45 minutes each [11].

#### 3.2.4. Main Results

In the study by Deen et al., patients showed a statistically significant post-intervention improvement in patient activation, as compared to the baseline measurement [12]. In the study by Hamann et al., as compared to the control group, the intervention group showed a statistically significant increase in participation, and responsibility for, decision-making regarding their treatment, which remained unchanged after six months [13]. Over time, the patients in the intervention group became more skeptical, both with regard to their treatment and with regard to their faith in their doctors and were classified as “more difficult” by their physicians [13]. However, even after six months, most patients in the intervention group stated they were still taking their medications [13]. The results in Alegria et al. showed that the DECIDE intervention led to a statistically significant improvement in patient activation and self-management in the intervention group compared to the control group. The authors also computed the effect size as a quantitative measure of the magnitude of a phenomenon (< 0.30 = small effect, <0.50 = medium effect and ≥0.50 = large effect) [27]. The results of the DECIDE intervention nearly reached a medium effect size for patient activation (d = 0.26) and for self-management (d = 0.22) [11]. 

#### 3.2.5. Study Quality

Together, the evidence for the effects of patient training in the selected and analyzed studies appears to be quite heterogeneous. Following evaluation according to the SURE criteria, one study had high internal validity [11], and two studies had distinct methodological limitations [12,13]: No control group was included in the before-and-after-study by Deen and colleagues. The study by Hamann et al. was only carried out at one center with a small sample, and therefore has low external. In addition, the studies by Hamann et al. and Alegria et al. dealt with patients in a psychiatric setting. It remains questionable as to what degree the results and the interventions can be transferred to patients with other conditions. It should be further noted that the authors of the analyzed studies made no statements concerning the relevance of their findings.

### 3.3. Studies on Continuing Medical Education

#### 3.3.1. Study Design and Objectives 

The 13 studies on continuing medical education for physicians comprised eight studies with a (cluster) randomized controlled design [16,19,20,21,22,23,24,25], three before-and-after-studies [14,17,26], one controlled-cohort study [18], and one study which implemented a mixed-methods design (combined qualitative and before-and-after-study designs) [15]. 

In five studies, the following target parameters were measured in patients:

One study measured patient participation in medical decisions using a questionnaire on shared decision-making (SDM-FB) [18].

Another study examined patient participation in (Patients’ Perceived Involvement in Care Scale, MSH-Scale), adherence to (internally-developed scale), and satisfaction (CSQ-8-questionnaire) with the treatment of depression [22].

One study examined whether participation (SDM-FB) increased due to intervention, and whether blood pressure values decreased compared to patients with a conventional antihypertensive therapy [25].

Two studies examined the effects of an intervention on, among other things, the quality of doctor-patient interactions (FAPI) and on the decision-making process (Satisfaction with Decision Scale, Decisional Conflict Scale) in patients with fibromyalgia [15,16].

In three studies, the following target parameters were examined for physicians:

One study used the Roter Interaction Analysis System to examine the extent to which patient-related communication improved [23].

One study used the Physician Satisfaction Questionnaire and a scale for physician-patient centeredness to examine whether an internally developed training program had an effect on the attitude and behavior of physicians toward patients with chronic pain during their treatment with opioids [24].

In another study, a training program was evaluated using, among other things, a multiple-choice test to examine whether skills in shared decision-making could be improved [17].

Five studies examine the following parameters in patients and physicians: 

One study examined whether a training course enabled physicians to present clear and ethically correct information on treatment options as well as to encourage patients to use shared decision-making [14]. The Decisional Conflict Scale was used for this purpose [14].

One study used a training program that should help to make decisions regarding the intake of antibiotics to treat acute respiratory infections [21]. An internally developed questionnaire was used [21].

Two studies used, among others, the D-Option Scale and a modified version of the Control Preference Scale to examine the extent to which a training program influenced decisions by physicians and patients regarding antibiotics intake to treat acute respiratory infections [19,20].

One study examined communication behavior of physicians and of patients using the Roter Interaction Analysis System [26].

#### 3.3.2. Number of Course Participants

Various study populations were considered in the studies on continuing medical education. In five studies, both physicians and patients were questioned regarding the impact of continuing medical education [15,16,19,20,21]. Six studies questioned either physicians [17,24] or patients [14,18,22,25]. There was no interview in two studies, which instead recorded physician consultations before and after the courses and analyzed the communication behavior using Roter Interaction Analysis System [23,26]. The number of physicians in the continuing education courses varied in the studies (see Table 1).

#### 3.3.3. Interventions

##### Persons Involved and Intervention Timing

In ten studies, the professions of the persons conducting the interventions were not identified [15,16,17,18,19,20,22,23,24,25]. In one study, the intervention was carried out by a communication trainer [26], and another indicated that the intervention was led by a clinical psychologist with experience in occupational skills training [14]. In addition, the study by Légaré et al. included comments that the research team was involved in the intervention [21]. None of the studies included information as to the point in time at which the intervention took place.

##### Description of the Interventions

The interventions were carried out differently in the various studies (see Table 3). In seven studies, the interventions were carried out as group interventions within the context of one [14,25] or more workshops [17,18,21,23,26]. The workshops included theoretical introductions, communication training, role playing, practical exercises, and video demonstrations. In addition to group interventions, six studies included additional elements, such as using decision-making aids or providing patient information and internet-based tutorials [15,16,19,20,22,24].

##### Duration

The duration of the continuing education courses varied in each study. In six studies, the duration varied between two and nine hours [14,17,18,24,25,26]. The first version of the program by Légaré et al. was three hours [21]. During its further development, the program was extended to four hours [19,20]. The intervention in Bieber et al. 2006 and Bieber et al. 2008 lasted a total of 18 hours [15,16]. In the study by Maatouk-Bürmann and colleagues, the intervention consisted of a three-day communication workshop [23]. Loh et al. reported that the training covered a period of six months and consisted of five sessions [22]. However, there was no information about the total duration of the intervention [22].

#### 3.3.4. Main Results 

Due to different measured parameters and outcomes and the heterogeneity in the studies, we can only summarize the results narratively in the following.

In a total of three studies, the intervention had no effect on the target parameters measured. The study by Hölzel et al. of insured patients detected no effects on inclusion in medical decisions or on other target parameters, such as quality of life [18]. Bernhard et al. found that there was no effect on patients’ decision-making confidence two weeks after the consultation [14]. In general, patients were satisfied with their treatment decisions and with their doctors’ capabilities [14]. The training course in the study by Tinsel et al. had no effects on patient involvement, changes in blood pressure values, adherence, or awareness [25].

Ten studies showed statistically significant and, in part, clinically relevant improvements in the target parameters following the intervention. The study by Timmermans found a significant improvement in the physicians’ communication behavior [26]: The physicians commented more frequently on psychosocial aspects and allowed more questions about diagnoses [26]. Following the training course, the patients in the intervention group were involved more closely than in the control group: They had the opportunity to address their conditions and concerns, expressed more about their ideas regarding their diagnoses and prognoses, and were more active in the decision-making process [26]. In the study by Maatouk-Bürmann et al., the physicians who participated in the intervention showed a significant and, according to the authors conclusions, clinically relevant improvement in patient orientation three months after the intervention [23]. The study by Sullivan et al. showed that the degree of patient orientation was the same in both groups three months following the intervention [24]. However, significantly more attention was paid to information transfer in the intervention group [24]. The physicians in the intervention group reported abiding by joint decisions more and reconsidering their treatment management of pain patients more often [24]. In the study by Loh et al., there was a statistically greater increase in patient involvement in the intervention group than in the control group [22]. There were no differences between the intervention group and those treated with conventional therapy with regard to adherence [22]. No effects on the degree of depression were found following the intervention [22]. The training course in the study by Bieber et al. was evaluated positively by the physicians who participated and, in their opinions, was effective in improving awareness of shared decision-making and interaction competencies [17]. In Bieber et al. 2006 and Bieber et al. 2008, the intervention had significant positive effects on the quality of the doctor-patient interaction [15,16]. In the studies by Légaré et al., the intervention led to significantly fewer decisions to use antibiotics [20,21]. In two studies, patients stated that they perceived having a more active role in the decision-making process following their physicians’ participation in a continuing education course [19,20].

#### 3.3.5. Study Quality

The validity (evidence) of the selected studies varies. Three studies exhibit intermediate internal validity [18,24,26]. In two studies, study participants were not randomly assigned to the treatment groups [18,26], the number of participants was too low [24], or the intervention was insufficiently described [18]. Ten studies exhibit high internal validity [14,15,16,17,19,20,21,22,23,25].

## 4. Discussion

The review looked at studies on patient education and continuing medical education that were carried out to promote shared decision-making. Three studies on patient education [11,12,13] and 13 studies on continuing medical education [14,15,16,17,18,19,20,21,22,23,24,25,26] were included. The studies were analyzed using various criteria.

The identified studies on patient education and on continuing medical education showed a heterogeneous picture for all criteria. It was clear from the outset that the studies pursued different target parameters. There were also large differences in the number of participants, which ranged from eight to 647 in the individual studies. The interventions in the studies were implemented differently. In six out of 13 studies on continuing medical education, the group interventions were combined with additional elements, such as additional patient information. In some cases, various professions were involved in implementing the intervention. The results from the studies on patient education showed a significant increase in patient activation. The studies on continuing medical education showed a more ambiguous picture: In three studies, which also included clinical target parameters, no statistically significant effects of the intervention could be measured [14,18,25]. This corresponds to results of a systematic review by Sanders and colleagues, who found no statistically significant changes in disease-related target parameters [28]. The other ten studies showed significant improvements in certain target parameters, for example in communication behavior and patient-centeredness. 

The studies reported little about satisfaction with the implemented patient education and continuing medical education courses. Four studies assessed and evaluated the interventions themselves [17,19,20,21]. The study by Bieber et al. evaluated the implemented training course directly with the physicians [17]. In the studies by Légaré et al., a continuing medical training program was systematically refined and improved [19,20,21]. The other studies did not conduct proximal evaluations of the assessments by the physicians being trained, nor were the interventions developed further based on those assessments. The implemented interventions have not been evaluated.

The present review has several methodological limitations. It is probable that conducting the literature search in two databases with the implemented inclusion and exclusion criteria was slightly limiting. It is possible that searching in additional databases would have resulted in more hits. Shared decision-making is also important during the treatment and care processes in other professions, and there are initial studies which examine educating shared decision-making in other professions [29,30]. For the patient education courses, only studies which aimed to promote communication skills within the scope of shared decision-making were included. Studies that examined certain disease-specific education programs or which had already implemented training manuals [31] were not included in the current review. The use of two reviewers during the selection process and during study content extraction should be emphasized as a strength of the present review.

## 5. Conclusions

The differences found in the studies made it difficult to compare the interventions and the results. The question of whether patient education and continuing medical education courses contribute to enhanced shared decision-making cannot be finally answered on the basis of the selected studies. In summary, there is a great need for more research in this area. We can presume that there is a current lack of studies which systematically evaluate patient education and continuing medical education courses as interventions. In the future, therefore, it makes sense for both areas to develop interventions for various settings and subsequently to evaluate and further develop these interventions.

## Figures and Tables

**Figure 1 ijerph-16-02482-f001:**
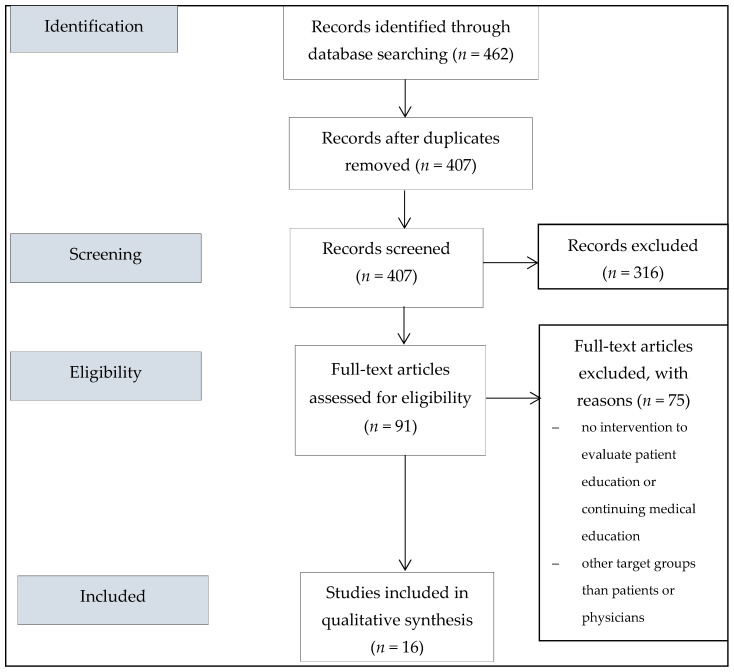
Study selection.

**Table 1 ijerph-16-02482-t001:** Study overview—Patient education and continuing medical education.

Literature	Topic	Country	Design	Characteristics of the Course Participants	
				*n*	Age
Alegria et al. 2014 [11]	PE	United States of America/Puerto Rico	Randomized controlled trial	647	18 to >65
Deen et al. 2011 [12]	PE	United States of America	Before-and-after-study	252	39 (Mean)
Hamann et al. 2011 [13]	PE	Germany	Randomized controlled trial	61	40,7 (Mean)
Bernhard et al. 2012 [14]	CME	Australia, New Zealand, Switzerland, Germany, Austria	Randomized controlled trial	62	39.5 (Mean)
Bieber et al. 2006 [15]	CME	Germany	Mixed Methods Study (qualitative and Before and After Study)	13	Not specified
Bieber et al. 2008 [16]	CME	Germany	Randomized controlled trial	10	30.8 (Mean)
Bieber et al. 2009 [17]	CME	Germany	Before-and-after-study	123	45 (Mean)
Hölzel et al. 2012 [18]	CME	Germany	Controlled cohort study with longitudinal design	33	Not specified
Légaré et al. 2013 [19]	CME	Canada	Randomized controlled trial	270	42.9 (Mean)
Légaré et al. 2012 [20]	CME	Canada	Randomized controlled trial	149	Not specified
Légaré et al. 2010 [21]	CME	Canada	Randomized controlled trial	33	Not specified
Loh et al. 2007 [22]	CME	Germany	Randomized controlled trial	42	33.77 (Mean)
Maatouk-Bürmann et al. 2016 [23]	CME	Germany	Randomized controlled trial	23	48.4 (Mean)
Sullivan et al. 2006 [24]	CME	United States of America	Randomized controlled trial	45	Not specified
Tinsel et al. 2013 [25]	CME	Germany	Randomized controlled trial	36	Not specified
Timmermans et al. 2006 [26]	CME	Netherlands	Before-and-after-study	8	33 (Mean)

Note: PE: Patient Education; CME: Continuing Medical Education.

**Table 2 ijerph-16-02482-t002:** Interventions—Studies on patient education.

Study	Type of Intervention	Applied Methodology	Brief Description of the Intervention	Duration
Alegria et al. 2014 [11]	Group intervention (DECIDE intervention)	Brainstorming, summary, role play exercises and practical exercises	- First training session: Patients are sensitized and encouraged to participate in decision-making in preparation for their role in physician consultations. - Second training session: The second session deals with treatment decisions with regard to role, processes, and reasons. - Third training session: The acquired skills are intensified and affirmed. Patients should identify additional sources, in order to answer questions.	30–45 min per training session with three training sessions total
Deen et al. 2011 [12]	Individual interviews	Interviews, brainstorming	- Understanding decisions: Thoughts on a recent decision and discussion on questions that should be considered to make a decision. - Choose a focus/topic relevant to current healthcare visit: The patients are asked whether they have questions for their physician with regard to the subsequent consultation. - Evaluate patient’s level of activation and brainstorming: Subsequent joint brainstorming to assist patients develop a catalogue of questions for their consultation with the physician. - Identify different types of questions: Sorting the questions according to open and closed questions (information about the purpose of open and closed questions): joint formulation of questions. - Prioritize questions: Assistance prioritizing the questions for the subsequent consultation and weighting them according to relevance. - List: After completing their questions, patients receive a list to use during their consultation.	Not specified
Hamann et al. 2011 [13]	Group intervention	Role play exercises	- Derived from theoretical considerations, adaption of related approaches and pilot testing the training. - Promotion of motivational and behaviour-oriented aspects. - Puts additional emphasis on mutual support and interaction.	Total of five hours, number of appointments not specified

Note: DECIDE: Decide the problem, Explore the questions, Closed or open-ended questions, Identify the who, why, or how of the problem, Direct questions to your health care professionals, Enjoy a shared solution.

**Table 3 ijerph-16-02482-t003:** Interventions—Studies on continuing medical education.

Literature	Type of Intervention	Applied Methodology	Brief Description of the Intervention	Duration
Bernhard et al. 2012 [14]	Group Intervention	Workshop with video modelling ideal behaviour and role play exercises	- Before the workshop: The participants must read the documents. - Workshop: Ensuring an SDM framework, structuring information, clear demonstration of various types of information, taking controversial information into account. - After the workshop: One month after the workshop, the participants are called again and encouraged to implement the learned behaviour. Furthermore, follow-up support by phone is offered within the two months following the workshop.	Total of seven hours and one session
Bieber et al. 2006 [15] Bieber et al. 2008 [16]	Mixed (Individual and Group Intervention)	Role play exercises, interactive talks, analysis of instructional videos with standardized patients	Training program - First module: Provision of a computer-based information tool for patients on the topic of “fibromyalgia”, consists of information on general symptoms, diagnosis, pathogenesis, treatment options, and prognosis is conveyed. - Second module: SDM communication training for physicians covers verbal and non-verbal communication, recognizing patient wishes, reflexion, appropriate responses, dealing with subjective perceptions of disease, and other emotional aspects.	Total of 18 hours, number of sessions not specified
Bieber et al. 2009 [17]	Group Intervention	Interactive presentations, model films on consultations, instructional videos with standardized patients, group discussion, practical exercises, role-playing of simulated consultations	Training program - First module: covers topics like patient preferences, theoretical framework of SDM, basic skills, effects, indications, limitations, and the pro and contra of SDM. - Second module: deals with uses of SDM in patient-centred communication, communication techniques, challenges with difficult patients, and the dynamics of doctor-patient-interaction.	Total of eight hours for two sessions
Hölzel et al. 2012 [18]	Group Intervention	Not described	Training program - First course: greater involvement of patients in decisions regarding therapy decisions and their implementation. - Two additional courses on doctor-patient-communication (focus: dealing with “difficult patients”).	Total of nine hours and three courses
Légaré et al. 2010 [21]	Group Intervention	DECISION + training program with video games, practical exercises, decision support tools, educational material	- Interactive workshops: advantages and disadvantages, as well as risks of various treatment options; techniques for risk communication; strategies for supporting patients in SDM. - Reminders of expected behaviours: Recalling SDM behaviour learned in the workshop, the benefit of decision aids, and information on current studies. - Feedback to physicians on the agreement between their decision conflict and that of their patients.	Total of three hours on three different days
Légaré et al. 2013 [19] Légaré et al. 2012 [20]	Mixed (Individual and Group Intervention)	DECISION + 2 training program with web-based self-tutorial, face-to-face, interactive sessions using videos, exercises and decision support tools	- Online self-tutorial: Introduction to the SDM process, diagnostics, treatment, effective communication of risks and benefits, promoting active patient participation. - Interactive workshop: Presenting diagnostic possibilities, using effective communication strategies, identifying patient preferences and values, involving patients in decisions, implementing decision aids in medical practice. - Decision support tool: available in the consultation office.	Total of four hours in one course
Loh et al. 2007 [22]	Group Intervention	Multi-faceted program with specialized lectures with accompanying questions, discussion rounds, facilitation practice, role-playing, video exemplars, vignettes	- Physician training: Physicians complete five training sessions on treating depression according to the standard medical guidelines; each training session consists of four modules. The modules should promote patient abilities, so that they can better participate in making medical decisions. - Decision board for use during the consultation. - Printed patient information	Not specified (five training sessions in six months)
Maatouk-Bürmann et al. 2016 [23]	Mixed (Individual and Group Intervention)	Role play exercises with fictitious patients, video feedback, direct informative feedback, and debriefing	Communication training - Theoretical introduction to specific communication models and interview techniques, - Teaching specific interview techniques, such as the WEMS technique (Waiting, Echoing, Mirroring, Summarizing) and the NURSE model (Naming, Understanding, Respecting, Supporting, Exploring), to improve patient-related communication.	Total of 24 hours on three different days and two hours for individual feedback
Sullivan et al. 2006 [24]	Mixed (Individual and Group Intervention)	Videos and discussion	Training program - In the first session, videos are used to practice simulated situations. The videos contain information on the SDM model, guiding a conversation, recommendations on SDM documentation, and recommendations for establishing treatment goals with patients. - In the second session, recorded video material is discussed with patients.	Total of two hours and two sessions
Tinsel et al. 2013 [25]	Group Intervention	Role play exercises, patient information in the form of flyers	Training program - Information about hypertension/risk communication and doctor-patient-communication. - Process steps for SDM/motivational interview techniques/introducing decision-making tables.	Total of six hours, number of sessions not specified
Timmermans et al. 2006 [26]	Group Intervention	Group discussion, training with simulation patients, video recording, individual feedback, a short written notice about the desired behaviour	Training program - Training in specific communicative behaviours. - Individual oral feedback. - A brief guideline as a reminder of the trained behaviours. - After each included consultation, physicians receive a checklist of the guideline-related behaviours.	Total of six hours in two sessions, and an additional three hours for individual feedback

Note: SDM: Share decision-making.

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
