# Peer review of "Patient Education and Continuing Medical Education to Promote Shared Decision-Making. A Systematic Literature Review"

_ijerph, 2019, doi:10.3390/ijerph16142482_

Reviewer 1 Report

In abstract: Suggestion to remove: “was carried out from the beginning of April through mid-June 2016.”

In line 65 to 67: suggested passing this information to the methods

In line 72 to 73: the boolean phrase has only one of its components, in this case the outcomes

Line 74: There is a need to better, systematizing information better: inclusion criteria and then the exclusion criteria.

Line 92: reference 9 refers us to an instrument: http://www.cardiff.ac.uk/insrv/libraries/sure/doc/SURE_RCT%20and%20other%20experiemental%20studies_Checklist_2013.pdf, not accessible and only for experimental studies.

In Materials and Methods: Data extraction should be referred to independently by two researchers.

In Figure 1: to remove “Publications before 2006”, this condition should be present in the inclusion criteria and the research (PRISMA flowchart) should only reflect the articles as of that date. The reasons for exclusion should be more explicit (n=?). What do you mean by: “no evaluation” or “other target groups”.

In Table 1: is missing the caption:RCT, USA. The column: Gender, does not add useful information, because it is not discussed in the results and is not specified in most studies.

In Table 2: could be added a column, with the methodology used in each study, with for example: brain storming, roll play, etc. Which, alias, are cited in the text.

In Table 3: could be added a column, with the methodology used in each study, with for example: methods for effective communication (interview technique, structuring of information, verbal and non-verbal communication,...), simulation, etc.

Overall, the data presented in Tables 2 and 3 could be more complete and systematized, as it would make it easier to read them.

Author Response

Dear Reviewer 1,

thank you very much for your valuable feedback. We carefully considered your comments and suggestions. The changes made in the manuscript are in blue. Please find below a point by point response:

In abstract

Suggestion to remove: “was carried out from the beginning of April through mid-June 2016.”

Answer: We changed the sentence to the following: We systematically searched for suitable studies in two databases (Pubmed and the Cochrane Database of Systematic Reviews) from the beginning of April through mid-June 2016 (Abstract, page 1, line 19 to 21)

In line 65 to 67: suggested passing this information to the methods

Answer: Thank you for this suggestion. We passed this information to the methods (page 3, line 94 to 96)

In line 72 to 73: the boolean phrase has only one of its components, in this case the outcomes

Answer: We agree. In order to find studies for our specific aim, we carried out a very broad search. So we only used the boolean phrase "AND".

We added the following sentences: In order to find studies for our specific research aim, we performed a broad-based search using the following keywords: “Patient participation” (MeSH) AND “Patient Education as Topic” (MeSH), “Patient participation” (MeSH) AND “training”, “Patient participation” AND “Patient Education” (page 2, line 70 to 74)

Line 74: There is a need to better, systematizing information better: inclusion criteria and then the exclusion criteria.

Answer: Thank you for the comment. We restructured this section and split the information into two sections: inclusion criteria and exclusion criteria (page 2, line 75 to 85)

Line 92: reference 9 refers us to an instrument: http://www.cardiff.ac.uk/insrv/libraries/sure/doc/SURE_RCT%20and%20other%20experiemental%20studies_Checklist_2013.pdf, not accessible and only for experimental studies.

Answer: Thank you for this comment. We checked the URL and added the accessible URL to the references. The instrument is for the critical appraisal of randomized controlled trials and other experimental studies, so we were able to use it well for the quality appraisal of the studies with an interventional design. For the cohort study with the longitudinal design (Hölzel et al. 2012), we conducted the critical appraisal again with the SURE checklist for cohort studies. We added the following sentences: For studies with an interventional design, we applied the SURE checklist for randomised controlled trials and other experimental studies [9]. For observational studies, we used the SURE checklist for cohort studies [10]. (page 3, line 97 to 100)

In Materials and Methods: Data extraction should be referred to independently by two researchers.

Answer: We agree that data extraction should be independently conducted by two researchers. We considered this during the review, so we changed the following sentence: The two reviewers (AW and NR) read the studies and extracted independently content of interest (page 3, line 92 to 93)

In Figure 1: to remove “Publications before 2006”, this condition should be present in the inclusion criteria and the research (PRISMA flowchart) should only reflect the articles as of that date. The reasons for exclusion should be more explicit (n=?). What do you mean by: “no evaluation” or “other target groups”.

Answer: We agree. We added “publication before 2006” to the section exclusion criteria. We added the following information to the flowchart: no intervention to evaluate patient education or continuing medical education; other target groups than patients or physicians (flowchart, page 4)

In Table 1: is missing the caption: RCT, USA. The column: Gender, does not add useful information, because it is not discussed in the results and is not specified in most studies.

Answer: Thank you. We corrected this in table 1. Concerning the column “Gender”, you are right. We deleted this column (table 1, page 4-5)

In Table 2: could be added a column, with the methodology used in each study, with for example: brain storming, roll play, etc. Which, alias, are cited in the text.

Answer: We agree. Corrected according to the suggestion. We added an additional column with this information (table 2, page 7-8)

In Table 3: could be added a column, with the methodology used in each study, with for example: methods for effective communication (interview technique, structuring of information, verbal and non-verbal communication,...), simulation, etc.

Answer: We agree. Corrected according to the suggestion. We added an additional column with this information (table 3, page 10-14)

Overall, the data presented in Tables 2 and 3 could be more complete and systematized, as it would make it easier to read them.

Answer: Thank you for this comment. We agree. We presented the data in table 2 and 3 in a more systematical way (see table 2, page 7-8; table 3, page 10-14)

Reviewer 2 Report

Interesting paper looking at an area in which not much research is available. The study does not examine a specific hypothesis, rather makes a detailed listing of all research available on patient education and continued medical education to promote shared decision making. It is a valuable study because it gives a nice overview of the limited publications in these field and the heterogeneously of the studies available. It also underlines that more publications should be prepared regarding the outcomes of continued medical education courses. Most likely now a lot of continued medical education is done, but the methodology and outcomes are seldom written up in a publication. 

Detailed comments: 

- line 16 add that the continued medical education looked at is to enhance shared decision making

- line 39: English correct - 'In order TO promote ... in clinical practice...' 

- line 46 - remove 'also'

- line 48-51: correct sentence - reads difficult

- line 166 - 'nearly reached a medium effect size' - is not clear to me what is meant with this.

 - line 246 Chapter Duration - suggest to make it much shorter. 

- line 256 Main results: Suggest to add an extra sentence underlining that due that the different studies were measuring different parameters to evaluate their success, it is difficult to describe real results. 

What you want to measure in your study is did the patient education or the continued medical education result in enhanced shared -decision making on the short-medium-longer time. This was impossible to see in most studies. 

Author Response

Dear Reviewer 2,

thank you very much for your valuable feedback. We carefully considered your comments and suggestions. The changes made in the manuscript are in blue. Please find below a point by point response:

Comments and Suggestions for Authors

Interesting paper looking at an area in which not much research is available. The study does not examine a specific hypothesis, rather makes a detailed listing of all research available on patient education and continued medical education to promote shared decision making. It is a valuable study because it gives a nice overview of the limited publications in these field and the heterogeneously of the studies available. It also underlines that more publications should be prepared regarding the outcomes of continued medical education courses. Most likely now a lot of continued medical education is done, but the methodology and outcomes are seldom written up in a publication. 

Answer: Thank you.

Detailed comments:

line 16 add that the continued medical education looked at is to enhance shared decision making

Answer: We added the following sentence: This review analyses intervention and evaluation studies on patient education and continuing medical education which aim to enhance shared decision-making (Abstract, page 1, line 15 to 17)

line 39: English correct - 'In order TO promote ... in clinical practice...

Answer: Thank you. We corrected this (page 1, line 40)

line 46 - remove 'also'

Answer: corrected

line 48-51: correct sentence - reads difficult

Answer: Thank you for this comment. We restructured this complicated sentence and suggest the following solution: A high level of communication skills is required of medical professionals in order to inquire about patients' preferences regarding participation in medical decisions. Physicians should be able to communicate disadvantages, risks, and uncertainty sensitively and neutrally to patients (page 2, line 49 to 52)

line 166 - 'nearly reached a medium effect size' - is not clear to me what is meant with this.

Answer: We added the following sentence to explain what is meant with effect size: The authors also computed the effect size as a quantitative measure of the magnitude of a phenomenon (< .30 =small effect, <.50=medium effect and ≥.50=large effect) [27]. The results of the DECIDE intervention nearly reached a medium effect size for patient activation (d=0.26) and for self-management (d=0.22) [11]. (page 8-9, line 180 to 184)

line 246 Chapter Duration - suggest to make it much shorter.

Answer: We shortened this chapter. Thank you for this suggestion (page 14, line 263 to 271)

line 256 Main results: Suggest to add an extra sentence underlining that due that the different studies were measuring different parameters to evaluate their success, it is difficult to describe real results.

Answer: Thank you for this suggestion. We added the following sentence: Due to different measured parameters and outcomes and the heterogeneity in the studies, we can only summarize the results narratively in the following.(page 15, line 273 to 274)

What you want to measure in your study is did the patient education or the continued medical education result in enhanced shared -decision making on the short-medium-longer time. This was impossible to see in most studies. The question of whether patient education and continuing medical education courses contribute to enhanced shared decision-making can not be answered on the basis of the selected studies.

Answer: Thank you for this comment. We added the following sentence to our conclusion: The question of whether patient education and continuing medical education courses contribute to enhanced shared decision-making can not be finally answered on the basis of the selected studies (page 16, line 352 to 354)

Round  2

Reviewer 1 Report

Very good work.

Congratulations.

Only change: Line 72 to 74 - To search, in this case, you must use the "OR", not "AND".